# Dynamics of Cyclooxygenase-1 Positive Microglia/Macrophage in the Retina of Pathological Model Mice as a Biomarker of the Retinal Inflammatory Diseases

**DOI:** 10.3390/ijms22073396

**Published:** 2021-03-25

**Authors:** Kenichi Makabe, Sunao Sugita, Yoko Futatsugi, Masayo Takahashi

**Affiliations:** 1Center for Biosystems Dynamics Research, Laboratory for Retinal Regeneration, RIKEN, 2-2-3 Minatojima-Minamimachi, Chuo-ku, Kobe 650-0047, Japan; makabe.k1@gmail.com (K.M.); yoko.futatsugi@riken.jp (Y.F.); retinalab@ml.riken.jp (M.T.); 2Department of Ophthalmology, Kobe Kaisei Hospital, 3-11-15 Shinoharakitamachi, Nada-ku, Kobe 657-0068, Japan

**Keywords:** microglia, cyclooxygenase-1 (COX-1), biomarker, functional imaging, retinal inflammation

## Abstract

In an intraocular inflammatory state, microglia residing in the retina become active and migrate inside the retina. In this study, we investigated whether cyclooxygenase-1 (COX-1) expressed by retinal microglia/macrophage can be a biomarker for the diagnosis of retinal diseases. COX-1 was immunopositive in microglia/macrophage and neutrophils, while COX-2 was immunopositive in astrocytes and neurons in the inner layer of normal retina. The number of COX-1 positive cells per section of the retinal tissue was 14 ± 2.8 (mean ± standard deviation) in normal mice, which showed significant increase in the lipopolysaccharide (LPS)-administrated model (62 ± 5.0, *p* = 8.7 × 10^−9^). In addition to microglia, we found neutrophils that were positive for COX-1. In the early stage of inflammation in the experimental autoimmune uveoretinitis (EAU), COX-1 positive cells, infiltrating from the ciliary body into the retinal outer nuclear layer, were observed. The number of infiltrating COX-1 positive cells correlated with the severity of EAU. Taken together, the increased number of COX-1 positive microglia/macrophage with morphological changes were observed in the retinas of retinal inflammatory disease models. This suggests that COX-1 can be a marker of disease-related activities of microglia/macrophage, which should be useful for the diagnosis of retinal diseases.

## 1. Introduction

Microglia are parenchymal tissue macrophages in the central nervous system (CNS). They are involved in immune responses, as well as in nerve generation and maintenance [1]. In pathological states of the CNS, microglia exert neuroprotective, phagocytotic, and neuropathic properties, and are involved in both positive and negative aspects of the pathology [2].

In brain diseases, diagnosis by the detection of pathologically active microglia using positron emission tomography (PET) has been attempted [3,4,5,6]. Microglia express cyclooxygenase-1 (COX-1) [7,8,9], one of the two isozymes of cyclooxygenase that synthesize prostaglandin from arachidonic acid on the cell membrane [10,11]. It has been suggested that probes that label COX-1 are useful for imaging microglia by PET [12,13,14,15].

Transgenic mice, in which microglia are visualized by GFP-labeled Iba1 [16], Cx3Cr1 [17], and CSF1R [18], have been used for research. However, in terms of clinical use, the safety of the probes targeting these molecules is unknown. On the other hand, since the probe for COX is prepared by non-steroidal anti-inflammatory drugs (NSAIDs), which are clinically used, it should be safe for human use.

The retina is a part of the CNS as the brain. It has been reported that retinal microglia express COX-1 in normal conditions [9]. However, the expression pattern of COX-1 in diseased retina is unknown, and whether COX-1 is useful for the diagnosis of ocular diseases has not been examined. Similar to brain diseases, retinal microglia show proliferation and migration in various ocular diseases, such as diabetic retinopathy [19], glaucoma [20,21], and retinitis pigmentosa [22,23,24], both in humans and rodents. Therefore, the detection of microglia may be useful for the diagnosis of these ocular diseases.

The purpose of this study is to investigate whether COX-1 can be a biomarker of retinal microglia for the diagnosis of ocular diseases. To achieve this goal, we analyzed the expression of COX-1 in retinal microglia of diseased model mice, both in vitro and in vivo. For brain disease, three types of animal models, in which COX enzymes convert arachidonic acid into inflammation mediators, have been reported: an *N*-methyl-D-aspartate (NMDA)-administration model [25], an LPS-administration model [26], and an autoimmune encephalomyelitis (EAE) model [27]. Therefore, we prepared retinal disease models that have the same mechanisms of disease development, namely an NMDA-administration model, an LPS-administration model, and an experimental autoimmune uveoretinitis (EAU) model. With these models, we analyzed the expression of COX in the retina to examine whether COX can be a biomarker for diagnosis.

## 2. Results

### 2.1. Expression of COX-1 in Normal Retina

To examine the localization of COX in normal retina, retinal sections prepared from wild type C57BL/6J were analyzed by immunohistochemistry. In normal retina, all Iba1^+^ cells had ramified morphology with long processes (Figure 1a,b), representing resting microglia as shown by little expression of CD68 (activated microglia/macrophage marker) (Appendix A). These Iba1^+^ cells were COX-1 positive (Figure 1a,b). Cells expressing NeuN (neuronal marker) in the retina did not coincide with COX-1 staining (Figure 1c). In the innermost layer of the retina, there were cells expressing glial fibrillary acidic protein (GFAP) (astrocyte/Müller cell marker), which did not colocalize with COX-1 staining (Figure 1d). No cells expressing Ly6G (neutrophil marker) were found in the retina in normal mice (Figure 1e). Thus, only Iba1^+^ cells were COX-1 positive in normal retina.

### 2.2. Expression of COX-2 in Normal Retina

Localization of COX-2 in normal retina was examined. GFAP^+^ cells (astrocytes or Müller cells) in the innermost layer of the retina coincided with COX-2 staining (Figure 2a,b). Müller cells represented by GFAP-positive vertical lines were not clearly COX-2 positive (Figure 2a). Neuronal marker (NeuN) positive cells were COX-2 positive (Figure 2c). Ramified microglia, represented by Iba1^+^ cells with processes, showed weak staining of COX-2 (Figure 2d). These results indicate COX-2 is not a specific marker for any of the cell types.

### 2.3. Expressions of COX-1 and COX-2 in Primary Cultured Retinal Microglia/Macrophage and Splenic Macrophage

To compare the expression of COX in retinal microglia/macrophage between a normal state and while under inflammatory stimulation, cultured retinal microglia/macrophage and splenic macrophages, with or without LPS-administration, were analyzed by flow cytometry (Figure 3). LPS stimulates COX-mediated conversion of arachidonic acid into prostaglandin [26]. When LPS was administered, both retinal microglia/macrophage and splenic macrophages showed an increased COX-1^+^ rate. In the case of retinal microglia/macrophage, the COX-1 positive rate was 62.8% and the COX-2 positive rate was 3.15% after PBS-administration, while the COX-1 positive rate was 82.4% and the COX-2 positive rate was 2.96% after LPS-administration (Figure 3a). In the case of splenic macrophage, the COX-1 positive rate was 51.5% and the COX-2 positive rate was 2.53% after PBS-administration, while the COX-1 positive rate was 75.0% and the COX-2 positive rate was 3.13% after LPS-administration (Figure 3b). Retinal microglia/macrophage tended to have a higher COX-1 positive rate than splenic macrophages (Figure 3c). Both retinal microglia/macrophage and splenic macrophage had low positive rate of COX-2 (Figure 3d).

### 2.4. Analysis of CD45 and COX-1 Expression in CD11b-Positive Cells Collected from the CNS

The CD11b/COX-1 positive cells in the retina could be either microglia or macrophages. To determine whether microglia collected from the CNS express COX-1, we freshly isolated CNS cells from mouse brain and purified CD11b-positive cells by cell sorting (Figure 4a), and then we stained with COX1 and CD45. Most of the CD11b-positive cells collected from the CNS were CD45-low to medium, and CD45-high was a small population. Both CD45-low to medium and CD45-high, that represent resting microglia and activated microglia/macrophages, respectively [28], expressed COX-1 (Figure 4b).

### 2.5. COX-1 in the Retina of N-Methyl-D-Aspartate (NMDA)-Administered Retinal Degeneration Model

Retinal neurons in the inner nuclear layer (INL) express NMDA-receptors. Hyperactivation of neurons by NMDA leads to apoptosis or necrosis, depending on the magnitude of the excitatory response, which, in the retina, causes loss of retinal ganglion cells [29]. In a brain disorder model, it has been suggested that upon NMDA uptake, COX is upregulated and converts arachidonic acid, a fatty acid enriched in the lipid bilayer of the nervous system, into inflammation mediators, such as prostaglandin [25]. We investigated the localization of COX-1 in the retina after four days of NMDA administration. COX-1^+^ staining coincided with Iba1^+^ cells (Figure 5a,b). The morphology of Iba1^+^ cells had changed into amoeboid form, with shorter processes compared to ramified microglia (Figure 5a,b, also see Figure 1b for normal condition), indicating these Iba1^+^ cells were activated microglia or infiltrating macrophages, which was further supported by their expression of CD68 (Appendix A). None of the NeuN^+^ cells expressed COX-1^+^ (Figure 5c). The number of COX-1^+^ cells had increased by NMDA administration (28.5 ± 2.6 per section) compared to normal conditions (14 ± 2.8 per section) (*p* = 3.9 × 10^−5^) (Figure 5d).

### 2.6. COX-1 in the Retina of Lipopolysaccharide (LPS)-Administered Ocular Inflammation Model

Injection of LPS into the brain is known to increase the metabolism of arachidonic acid that is converted into pro-inflammatory lipid mediators by COX [26]. In order to know how COX-1 expression in the retina changes with the inflammatory state, the retina of the LPS-administered ocular inflammation model was examined by immunohistochemistry. Four days after the intraocular administration of LPS, numerous Iba1^+^ cells were accumulated in the inner layer of the retina, and COX-1^+^ staining coincided with Iba1^+^ cells (Figure 6a). These Iba1^+^ cells were amoeboid cells (Figure 6a), indicating they were activated microglia or infiltrated macrophages. Five days after the administration, Iba1^+^ cells scattered in all layers of the retina (Figure 6a). These Iba1^+^ cells at day five were also amoeboid cells (Figure 6a), which were CD68 positive (Appendix A), indicating they were activated microglia or infiltrating macrophages. At this time, Iba1^−^COX-1^+^ cells also scattered into the retina with the emergence of Ly6G (neutrophil marker) positive round cells that coincided with COX-1^+^ staining (Figure 6b). GFAP staining was enhanced compared to the normal retina (also see Figure 1d for normal condition), but no coincidence with COX-1 staining was found (Figure 6c). The number of COX-1^+^ cells had increased by LPS-administration (62 ± 5.0 per section) compared to normal condition (14.0 ± 2.8 per section) (*p* = 8.7 × 10^−9^) (Figure 6d).

### 2.7. Early Stage Inflammation in Experimental Autoimmune Uveoretinitis (EAU) Evaluated by COX-1^+^ Cells

It has been shown that COX is expressed in EAE [27]; that is, a mouse model of multiple sclerosis, an autoimmune disease targeting the myelin, which is made by immunizing the mice with myelin proteins. In the same manner, we generated an EAU model by immunizing the mice with a retinal protein hIRBP, which shows an autoimmune disease that targets neural retina and related tissues [30]. In EAU mice, floating cells were observed in the vitreous body by OCT imaging, which emerged nine days after immunization and increased over time (Figure 7a). In flat-mount retina of EAU, Iba1^+^ cells infiltrating into the ONL of the peripheral retina from the choroidal side increased during the observation. In the control group, Iba1^+^COX-1^+^ cells were found in the choroid but not on the retinal side. (Figure 7b). Iba1^+^ cells in the control mice and in EAU mice five days after immunization (day 5) had small a cell body and thin processes (Figure 7b). Iba1^+^ cells in EAU mice on days 7, 9, and 12 had thick processes that were clearly visible (Figure 7b), indicating they were slightly activated. Iba1^+^ cells on day 14 became amoeboid cells (Figure 7b) that were CD68 positive (Appendix A), indicating they were activated microglia or infiltrating macrophages. All Iba1^+^ cells coincided with COX-1^+^ staining (Figure 7b). The number of infiltrating COX-1^+^ cells into the ONL correlated with the in vivo OCT inflammation score. (Spearman *r* = 0.73, *p* = 0.012) (Figure 7c).

Taken together, the in vitro and in vivo data from three types of retinal disease models demonstrate that COX-1 can be a biomarker of retinal microglia/macrophage, which is useful for the diagnosis of ocular diseases.

## 3. Discussion

We verified whether COX-1 can be a marker for the diagnosis of inflammatory ocular diseases. Flow cytometric examination and histopathologic examination of retinal tissues in normal mice and inflammatory disease model mice showed that microglia/macrophage of the retina expressed COX-1, suggesting that COX-1 could be useful for tracing the dynamics of microglia/macrophage in inflammatory ocular diseases.

COX-1 was detected in cultured retinal microglia/macrophage, regardless of the administration of LPS, while COX-2 was hardly detected. A previous report showed increases in COX-2 expression in cultured microglia 8 h after stimulation with LPS [7]. Since we examined the COX expression after 72 h of LPS-administration, it could be possible that COX-2 was transiently expressed in response to LPS before our observation. If COX-2 is expressed only for a short period, it is difficult to capture and may not be suitable for clinical examination as a marker of inflammation. On the other hand, COX-1 was detected in retinal microglia/macrophage regardless of LPS stimulation, and should thus be a useful retinal microglial/macrophage marker.

It has been reported that over 98% of the CD11b positive cells collected and cultured from the retina by a similar method to ours are CD45 weak positive [31]. This suggests that most of the CD11b positive cells collected and cultured from the retina are microglia. Our analysis on CD45 expression in CD11b positive cells collected from mouse brain consistently showed that, in most cells, CD45 expression levels were medium to low, suggesting microglia were predominant in the CD11b positive cells collected from the CNS. Since COX-1 was also positive in the cells with higher expression levels of CD45, it was suggested that both resting microglia and activated microglia/macrophage are positive for COX-1.

In immunohistochemistry of normal mouse retinal tissue, the staining of COX-1 coincided with Iba1^+^ cells. Although previous studies reported that COX-1 was also expressed in amacrine cells and retinal ganglion cells [9], our results showed that the expression of COX-1 was relatively high in microglia/macrophage, suggesting COX-1 can be a biomarker of microglia/macrophage in retina. On the other hand, COX-2 was weakly positive in microglia, and was also positive in the inner layer of the retina (i.e., from the retinal nerve fiber layer (RNFL) to the inner plexiform layer (IPL)), and in some neurons in the INL. NeuN is strongly expressed in amacrine cells in the GCL and weakly expressed in amacrine cells and horizontal cells in the INL in mice [32]. Amacrine cells were reported to be COX-2 positive in mouse retina, while horizontal cells were negative [9]. Therefore, the NeuN^+^COX-2^+^ cells in the INL are considered to be amacrine cells. COX-2 has been reported to be negative in Müller cells of the retina [9], which is consistent with our results. In our study, the COX-2 positive area in the INL did not clearly coincide with GFAP staining. COX-2 was expressed ubiquitously in the inner retina and could not be a marker of a specific cell type.

In the NMDA-administration model, neurons undergo apoptosis upon the binding of NMDA to the NMDA-type glutamate receptors, resulting in excessive inflow of calcium ions into the neurons. Glutamate toxicity via the NMDA-type receptor in the retina mainly induces neuronal cell death in the ganglion cell layer (GCL) and the INL [33]. Glutamate toxicity is thought to be involved in retinal disorders of glaucoma [34] and diabetes [35]. In the retina of the NMDA-administration model, the inner layer of the retina is mainly lesioned [36]. In our immunostaining, COX-1 positive microglia changed their morphologies into an activated amoeboid form at the inner layer of the retina, which coincided with the major injury site of the NMDA-administration model. There are two possible mechanisms for microglial proliferation and activation in the NMDA retinal neurotoxicity model. First, agonists of ionotropic glutamate receptors, including NMDA, may induce activation and proliferation of microglia directly, which precedes neuronal cell death [37]. Second, ATP released from damaged neurons may stimulate microglia via the P2X7 receptor, one of the ion channel-linked ATP receptors, leading to the proliferation and migration of microglia towards the damaged site [38].

LPS is a cell membrane component of Gram-negative bacteria. Administration of LPS to the central nervous system stimulates arachidonic acid metabolism that causes neuroinflammation [26]. Moreover, LPS causes activation of microglia via toll-like receptor 4 [39]. In our experiments, probably intraocularly administrated, LPS infiltrated from the vitreous cavity into the inner layer and further into the outer layer of the retina, causing inflammation. Microglia/macrophage proliferation also spreads from the inner layer to all layers of the retina. With severe intraocular inflammation after LPS administration, we could observe a dramatic increase in COX-1 positive microglia/macrophage. We also observed COX-1 positive neutrophils, indicated by Ly6G^+^COX-1^+^ immunostaining. Therefore, two types of cells, microglia/macrophages and neutrophils, seem to be involved in the proliferation of COX-1^+^ cells in the LPS-treated ocular inflammation model. LPS administration causes the proliferation of microglia in vitro, which is affected by the concentration of TNFα around the microglia [40]. In vivo study showed an increase in TNFα in the eye after LPS administration [41]. In the LPS-administrated ocular inflammation model, neutrophils and macrophages infiltrate into the eye [42], which, together with our results, makes it reasonable to consider that these infiltrated neutrophils and macrophages may contribute to an increase in COX-1^+^ cells. Although COX-1 expression in neutrophils has not been reported, our ocular inflammation model showed COX-1^+^ neutrophils infiltrating into the retina, which suggests COX-1 may be an indicator of innate immune activity that underlies neuroinflammation. Although histologically neutrophils that have multinucleated spherical morphology differ from monocytes/macrophages and microglia, it might be difficult to distinguish between these cell types with non-invasive diagnostic imaging using COX-1, as clinical evaluation of cell types requires the collection of aqueous or vitreous humor.

In the early phase of EAU mice, there was a correlation between the number of Iba1^+^COX-1^+^ cells found in the ONL layer of the peripheral retina and the OCT score for inflammation. It is known that, in early stages of EAU, microglia migrate into the ONL and become active, followed by the infiltration of macrophages into the retina [43]. However, little is known about the exact mechanism underlying microglial activation and their migration into the ONL. In the photoreceptors of EAU mice, docosahexaenoic acid (22:6), the major membrane fatty acid of the photoreceptor, is peroxidized to form hydroperoxide (22:6 HP) [44]. Since 22:6 HP induces chemotaxis of phagocytic cells and microglia, 22:6 HP is considered to play a role in the migration of microglia into the ONL in the retina of EAU [43]. In recent years, ultra-wide-field image system has beewidely used in clinical practice, which enables the evaluation of the peripheral retina in humans. As leakage from peripheral retinal blood vessels provides an indication for the evaluation and management of anterior uveitis [45] and tuberculous posterior uveitis [46], the importance of evaluating the peripheral retina in uveitis has been recognized. Recently, we showed Iba1-positive cells (microglia/macrophages) invading into the retina and the choroid after retinal pigment epithelial cell–transplantation in mice [47,48,49]. Moreover, after transplantation of retinal pigment epithelial cell allografts in patients with age-related macular degeneration, an increase in CD11b-positive cells (monocytes/macrophages) in the blood was shown; one patient suffered from immune rejection [50]. However, we still do not have direct evidence for the involvement of retinal microglia and choroidal macrophages in ocular rejection. More recently, our study using *rd10* retinitis pigmentosa mouse models suggested that the accumulation of active retinal microglia might correlate with the degree of photoreceptor apoptosis, and the rate of outer nuclear layer thinning, in patients with retinitis pigmentosa [51]. If it becomes possible, in the future, to evaluate the retina by molecular imaging of microglia/macrophages, it may be useful for the diagnosis of uveitis, immune rejections after retinal transplantation, and retinal degeneration.

In the current study, we have not been able to examine COX-1 by PET because the resolution of PET is not sufficient for photographing the retina of mice at present (data not shown). However, the present study is potentially important, as an examination of the retina with a COX-1 probe in humans should be possible in the future. PET imaging devices are improving their resolutions [52], and studies of PET used for human retina have also been reported [53]. Several probes conjugated with fluorescent dyes have been developed for the detection of COX [54,55]. For instance, the detection of COX-2 during choroidal neovascularization with in vivo fluorescence fundus photography was shown in an animal model [56]. A separate in vitro study showed that a fluorescent probe of COX-1 was able to indicate human ovarian cancer cells OVCAR-3 which expressed COX-1 [57]. Therefore, the detection of COX-1 expressing retinal microglia in ocular diseases with fluorescent probes is becoming realistic for clinical use.

For clinical use, quantitative evaluation of COX-1 concentration should be useful as a biomarker. Generally, standardized uptake value (SUV) is used for quantitative evaluation of the target molecule in clinical PET imaging. For example, therapeutic effect against solid tumor could be judged by the rate of change in SUV before and after treatment [58]. As direct measurement of COX-1 concentration in patients is not clinically possible, quantitative evaluation by SUV might be useful.

## 4. Material and Methods

### 4.1. Isolation and Culture of Retinal Microglia

C57BL/6J mice were sacrificed at postnatal day 6, and their eyes were extracted and placed in 17.8 mL of ice-cold HBSS (Thermo Fisher Scientific, Waltham, MA, USA) with 200 μL of penicillin/streptomycin (Thermo Fisher Scientific). The corneas and lenses were removed from eyes, and retinas were carefully dissected from the eyecups under a microscope. Eight retinas were collected and shredded in a culture medium. The tissues were then treated with 5 μg/mL of 2.5% trypsin-EDTA (Thermo Fisher Scientific) and incubated for 30 min at 37 °C 5%CO_2_, stirring every 5 min. DMEM/F12 + Glutamax containing 10% fetal bovine serum (FBS) (Biological industries, Beit-Haemek, Israel) and penicillin/streptomycin was added for 1 h at 37 °C. The mixed retinal single-cell suspension was washed in DMEM/F12 + Glutamax and filtered through a 70 µm cell strainer (BD Falcon, Biosciences, Oxford, UK). The retinal single-cell suspension was seeded on a 10 cm culture dish (Nunc Thermo Scientific, Leicestershire, UK) in DMEM/F12 + Glutamax containing 20% FBS and penicillin/streptomycin. F12 + Glutamax was used to increase media stability, reduce ammonium, and provide more stable glutamine. After 10 days of culture, nonadherent cells were removed. The remaining adherent cells (i.e., microglial cells) were left to replicate further (2–3 weeks) in the culture dish. Cell morphology was observed by phase–contrast microscopy. Images were captured using a microscope (IMT-2; Olympus, Tokyo, Japan) equipped with a digital camera (Jenoptik Laser, Optik System GmbH, Jena, Germany). Images were processed using ProgRes C14 Imaging software (Jenoptik Laser). The purity of the cells collected and cultured from the retina, in a similar manner, has been reported; more than 98% of the cultured cells were positive for CD11b without high levels of CD45 expression, indicating the cells are microglia [31].

### 4.2. Isolation and Preparation of Splenic Macrophage

C57BL/6J mice were sacrificed and their spleens were extracted. Splenocytes were prepared from spleen with a 100 μm cell strainer, as per a previous report [59]. The purity of the primary cultured mouse splenic macrophage was reported to be as high as 80% by day 5 and >90% by day 7 of culture [60].

### 4.3. Isolation of CNS Microglia/Macrophage

Mouse brain dissected from 6 ICR mice at postnatal day 2 was digested in a papain solution (papain 165U, 1N NaOH 20 µL, L-cystein 2.4 mg, DNaseI 2 mg in DPBS 10 mL) at 37 °C for 30 min. After stopping the digestion with 30 mL of DMEM 10%FBS, cell suspension was obtained by passing through a 18G-needle and 70 μm cell strainer, centrifuged for 5 min at 300 *g* and suspended in 300 μL of RPMI 2%FBS. Cells were then blocked with CD16/32 Mouse BD Fc Block (BD Bioscience: clone 2.4G2) at 4 °C for 15 min, washed once, incubated with APC conjugated anti-CD11b Ab (Miltenyi Biotec: 130-091-241) at 4 °C for 30 min, washed twice, and then sorted with a BD FACS AriaII cell sorter (Becton, Dickinson and Company, Franklin Lakes, NJ, USA).

### 4.4. Flow Cytometry

Expressions of COX-1 and COX-2 in retinal microglia, splenic macrophages, and CD11b-positive CNS cells were examined by a FACS analysis. The cells were blocked with a CD16/32 Mouse BD Fc Block (BD Bioscience: clone 2.4G2) at 4 °C for 15 min, then stained with APC conjugated anti-CD11b Ab (Miltenyi Biotec: 130-091-241), PE conjugated anti-COX-1Ab (SantaCruz, Dallas, TX, USA: sc-1754 PE), PE conjugated goat anti-COX-2 Ab (SantaCruz: sc-1745 PE), FITC conjugated rat anti-CD45 (BioLegend: 30-F11), or isotype controls (goat IgG) at 4 °C for 30 min. All samples were analyzed on a FACS CantoII flow cytometer (Becton, Dickinson and Company, Franklin Lakes, NJ, USA), and data were analyzed by FlowJo software. From the FSC-SSC plot, fractions of macrophages and microglia were gated, and then CD11b positive cells among them were gated. Among the gated CD11b positive cells, the rate of COX-1 or COX-2 positive cells was analyzed. A FACS analysis was performed on the cells from the LPS-administration model 72 h after LPS (1 μg/mL)- or PBS-administration, and on CD11b positive cells isolated from mouse brain as described in Section 4.3.

### 4.5. Experimental Animals

C57BL/6J (B6J) mice were obtained from CLEA Japan, and they were housed at a local animal facility under standard laboratory conditions (18–23 °C, 40–65% humidity, and a 12 h light–dark cycle), with free access to food and water throughout the experimental period.

### 4.6. NMDA-Administration Retinopathy Model

Mice were anesthetized by an intraperitoneal injection of ketamine 120 mg/kg and xylazine 10 mg/kg. The pupils of the mice were dilated with 0.1% Phenylephrin and 0.1% Tropicamid. For the NMDA model, a 33-gauge bevel needle was connected to a Hamilton syringe (Sigma-Aldrich, St. Louis, MO, USA) and 2 μL of 10 mM NMDA (Sigma-Aldrich) was administered with a vitreous injection. Control mice were injected with 2 μL of PBS intravitreally.

### 4.7. LPS-Administered Ocular Inflammation Induction Model

Mice were anesthetized by an intraperitoneal injection of ketamine 120 mg/kg and xylazine 10 mg/kg. The pupils of the mice were dilated with 0.1% Phenylephrin and 0.1% Tropicamid. For the LPS model, a 33-gauge bevel needle was connected to a Hamilton syringe (Sigma-Aldrich) and 2 μg/2 μL of LPS from *E. coli* O26 (WAKO) was administered with a vitreous injection. Control mice were injected with 2 μL of PBS intravitreally.

### 4.8. Experimental Autoimmune Uveoretinitis Model

EAU was induced in 6-to-8-week-old female C57BL/6JJcl mice. To induce EAU, mice were immunized subcutaneously in the neck region with an emulsion containing 200 μg of human interphotoreceptor retinoid-binding protein peptide (hIRBP_1–20_, Eurofins Genomics, Tokyo, Japan) and *Mycobacterium tuberculosis* strain H37Ra (Difco, Detroit, MI, USA) in complete Freund’s adjuvant (Becton, Dickinson and Company), and injected intraperitoneally (i.p.) with 100 ng of pertussis toxin (Sigma-Aldrich) as an additional adjuvant.

### 4.9. Optical Coherence Tomography (OCT) Imaging

The pupils of the mice were dilated with 0.1% Phenylephrin and 0.1% Tropicamid before imaging. Artificial tears were used throughout the procedure to maintain corneal clarity. Sedation was induced by 1% isoflurane, delivered by a nose cone while the animal was seated in the Bioptigen AIM-RAS holder. SD-OCT images were obtained using a Envisu R2200-HR SD-OCT device (Bioptigen, Durham, NC, USA). We scored the severity of EAU according to the previously reported OCT score evaluation method for mouse EAU [61].

### 4.10. Immunohistochemistry

The mice were sacrificed after 18 h or 4 days of administration in the NMDA-administration model, after 5 days or 2 weeks of administration in the LPS-administration model, and after 5, 7, 9, 12, or 14 days of immunization in the EAU model. The eyeballs were enucleated and fixed with 4% paraformaldehyde. Retinas were peeled from the eyeball to prepare flat-mount specimens. Eyeballs for sectioning were embedded in an OCT compound (Sakura Finetek, Japan, Tokyo) and blocks were frozen at −80 °C. Sections were sliced in 10 μm thickness using an HM 560 CryoStar cryostat (Thermo Fisher Scientific, Waltham, MA, USA). Specimens were blocked with 10% donkey serum in 1 × PBS with 0.3% Triton X-100 for 1hour at room temperature, and then incubated with primary antibodies in 1 × PBS with 0.3% Triton X-100 for 24 h (retinal section), or for 5 days (retinal flat mount) at 4 °C. Primary antibodies used were rabbit anti-ionized calcium binding adaptor molecule 1 (Iba-1) (Wako, Osaka, Japan, 1:1000), goat anti-COX-1 (SantaCruz, 1:100), goat anti-COX-2 (SantaCruz, 1:100), rat anti-Ly6G (BioLegend, San Diego, CA, USA, 1:100), rabbit anti-NeuN (Abcam, Cambridge, UK, 1:500), and rabbit anti-GFAP (Dako Cytomation, Santa Clara, CA, USA, 1:200). After washing in 1 × PBS with 0.3% Triton X-100, retinas were incubated for 1 hour (section) or overnight (flat mount) with secondary antibodies: Alexa Fluor 488-conjugated anti-rabbit (Thermo Fisher Scientific, 1:1000), Alexa Fluor 546-conjugated anti-goat (Thermo Fisher Scientific, 1:500), or Alexa Fluor 647-conjugated anti-rat (Jackson immune research laboratory, West Grove, PA, USA 1:200) with DAPI (1:1500 Sigma-Aldrich). Images were acquired with a confocal microscope (LSM700, Zeiss, Oberkochen, Germany).

### 4.11. Statistical Analyses

For statistical analyses, commercially available software packages, Excel (Microsoft, Redmond, WA, USA) and Statcel3 (Add-in forms on Excel, H. Yanai, OMS, Tokyo, Japan) were used. *p* values less than 0.05 were evaluated as statistically significant.

## 5. Conclusions

In the current study, we demonstrated that retinal microglia/macrophage showed characteristic dynamics for each disease model, which we were able to monitor by COX-1 imaging. An increased number of COX-1 positive microglia/macrophage with morphological changes were observed in the retinas of retinal inflammatory disease model mice. In the future, when probes for PET or other devices are developed, COX-1 may be a useful biomarker of neuroinflammation in retinal diseases.

## Figures and Tables

**Figure 1 ijms-22-03396-f001:**
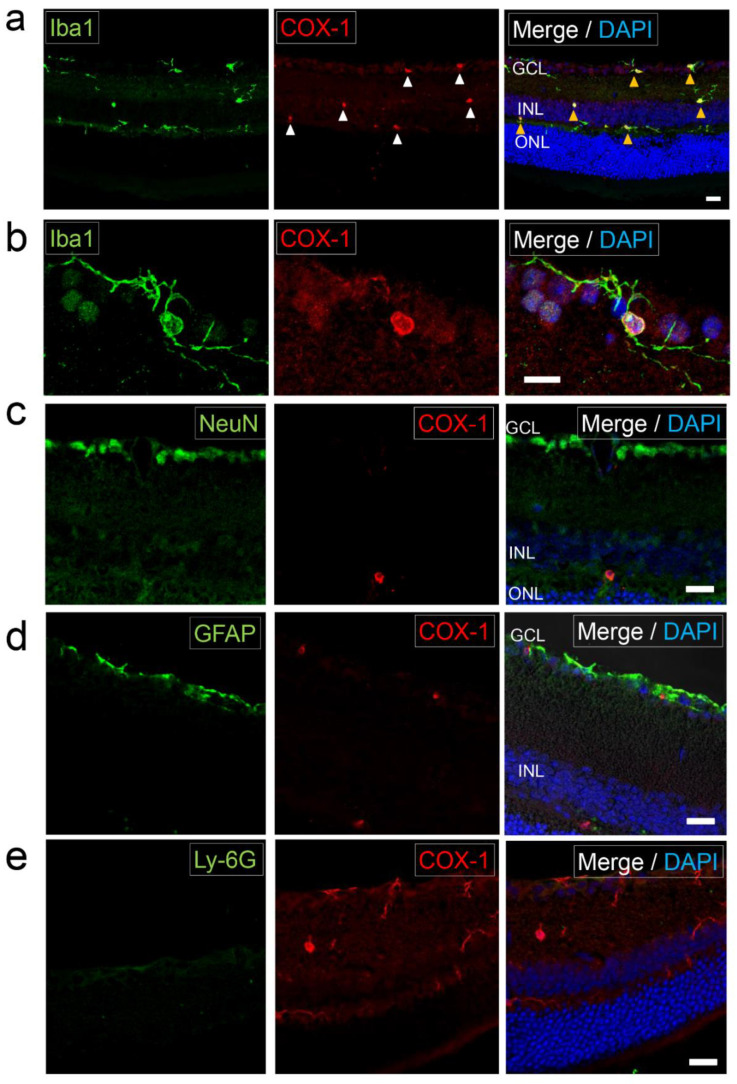
Expression of cyclooxygenase-1 (COX-1) in normal retina Photomicrographs of the retina in normal C57BL/6J mice. (**a**) The retinal section labeled with Iba1 (microglia/macrophage marker) and COX-1. There were COX-1^+^ cells in the GCL, IPL, INL, and OPL (white arrowheads). The COX-1 staining coincided with Iba1^+^ microglia (yellow arrowhead) that had ramified (resting state) morphology. Scale bar, 20 μm. (**b**) Photomicrographs showing a magnified image of microglia in the retina that had ramified morphology with long processes. Microglia with long processes were positive for Iba1 and COX-1. Scale bar, 10 μm. (**c**) The retinal section labeled with NeuN (neuronal marker) and COX-1. COX-1 staining did not coincide with NeuN^+^ cells. Scale bar, 20 μm. (**d**) The retinal section labeled with GFAP (astrocyte/Müller cell marker) and COX-1. COX-1 staining did not coincide with GFAP staining. Scale bar, 20 μm. (**e**) Retinal section labeled with Ly-6G (neutrophil marker) and COX-1. There were no Ly-6G^+^ cells in normal retina. Scale bar, 20 μm.

**Figure 2 ijms-22-03396-f002:**
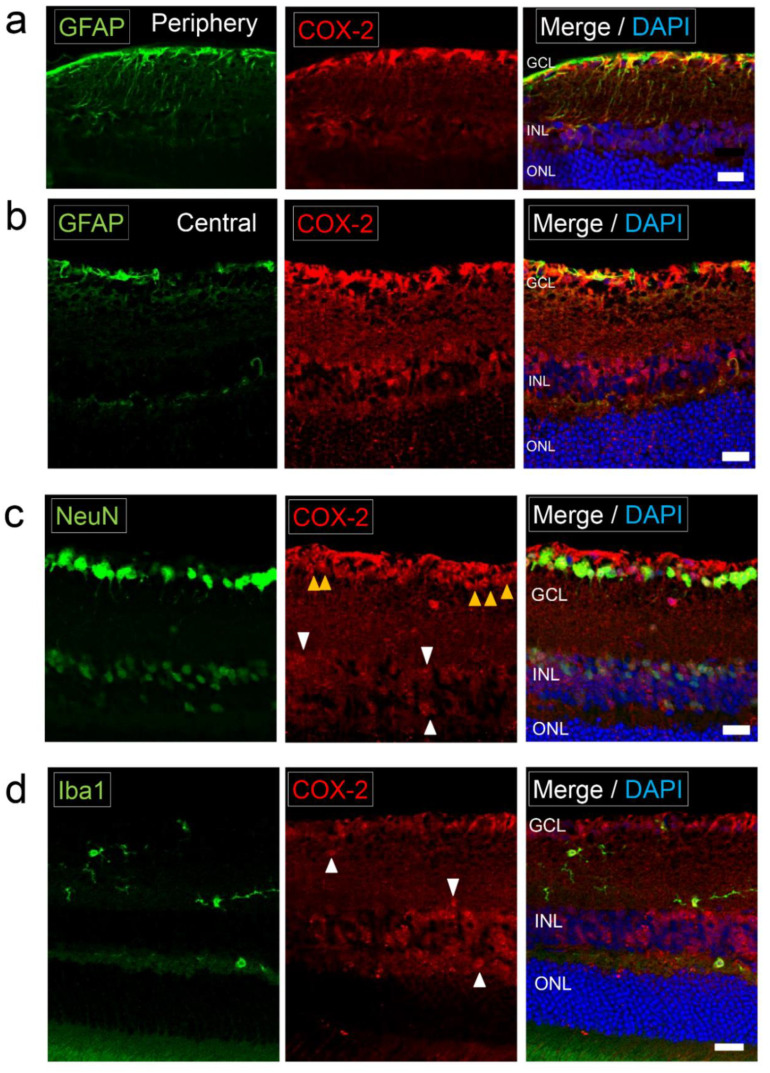
Expression of COX-2 in normal retina. Photomicrographs of the retina in normal C57BL/6J mice (**a**) The retinal section of the peripheral area labeled with GFAP (astrocyte/Müller cell marker) and COX-2. COX-2 was positive in GFAP^+^ cells. GFAP-positive vertical lines (Müller cells) were not clearly COX-2 positive. Scale bar, 20 μm. (**b**) The retinal section of the central area labeled with GFAP and COX-2. COX-2 staining coincided with GFAP staining. Scale bar, 20 μm. (**c**) The retinal section labeled with NeuN (neuronal marker) and COX-2. COX-2 was positive in NeuN^+^ neurons in the GCL (yellow arrowheads) and the INL (white arrowheads). Scale bar, 20 μm. (**d**) The retinal section labeled with Iba1 and COX-2. Iba1^+^ cells were ramified microglia with processes. COX-2 was weak positive in Iba1^+^ ramified microglia (white arrowheads). Scale bar, 20 μm.

**Figure 3 ijms-22-03396-f003:**
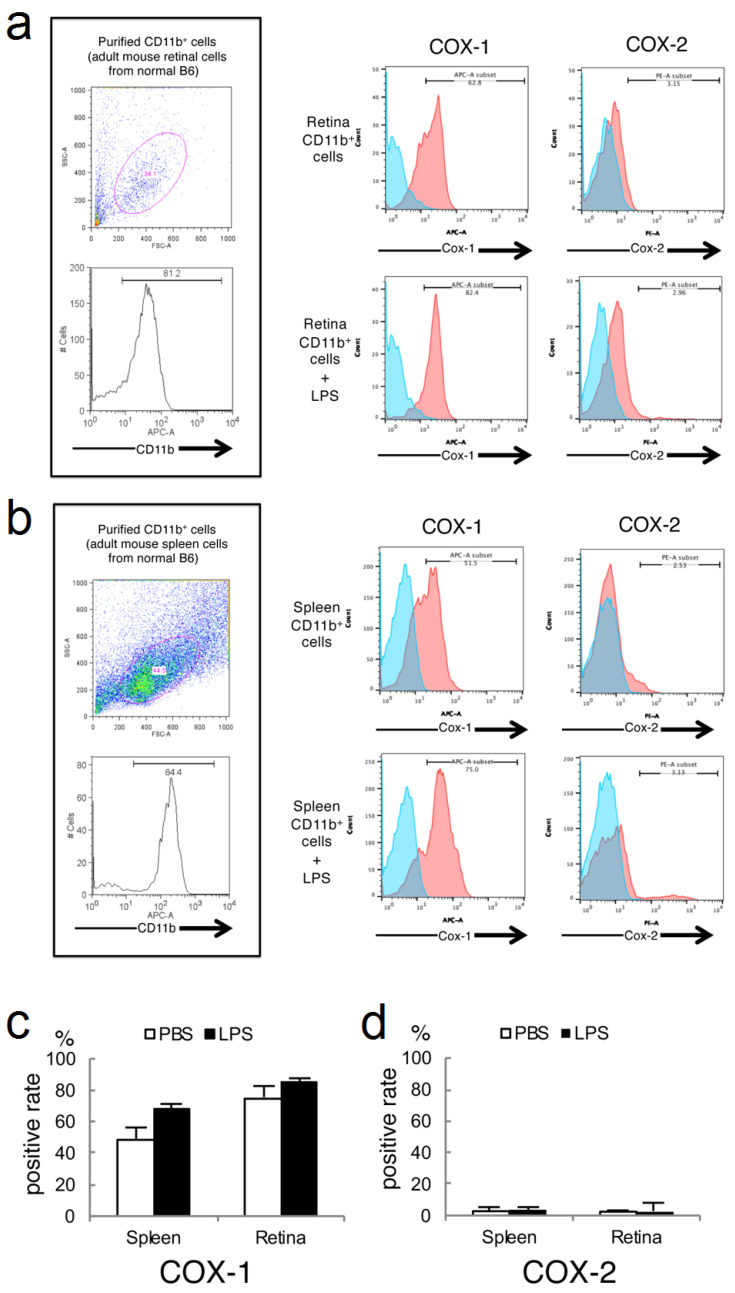
In vitro analysis of the expression of COX-1 and COX-2 in retinal microglia/macrophage and splenic macrophage. (**a**) Cultured retinal glia cells of C57BL/6J mice were stained with anti-CD11b, anti-COX-1, and anti-COX2. A FACS cytometric analysis was performed 72 h after PBS- (upper) or LPS- (lower) administration. Red plots show the results of COX-1 or COX-2 positivity. Blue plots show the results of isotype control. Numbers (%) in the histogram indicate the ratio of positive cells. Thresholds were set by the staining with isotype control of the antibody. (**b**) Cultured splenic cells of C57BL/6J mice were stained with anti-CD11b, anti-COX-1, and anti-COX2. PBS- (upper) or LPS- (lower) administrated samples were analyzed on a FACS flow cytometer. Red plots show the results of COX-1 or COX-2 positivity. Blue plots show the results of isotype control. Numbers (%) in the histogram indicate the ratio of positive cells. Thresholds were set by the staining with isotype control of the antibody. (**c**) A graph showing the positive rate of COX-1 in splenic macrophages and retinal microglia/macrophage after administration of PBS (white bar) or LPS (black bar). An error bar showing standard deviation (SD). (*n* = 3). (**d**) A graph showing the positive rate of COX-2 in splenic macrophages and retinal microglia/macrophage after administration of PBS or LPS. Error bar shows SD. (*n* = 3).

**Figure 4 ijms-22-03396-f004:**
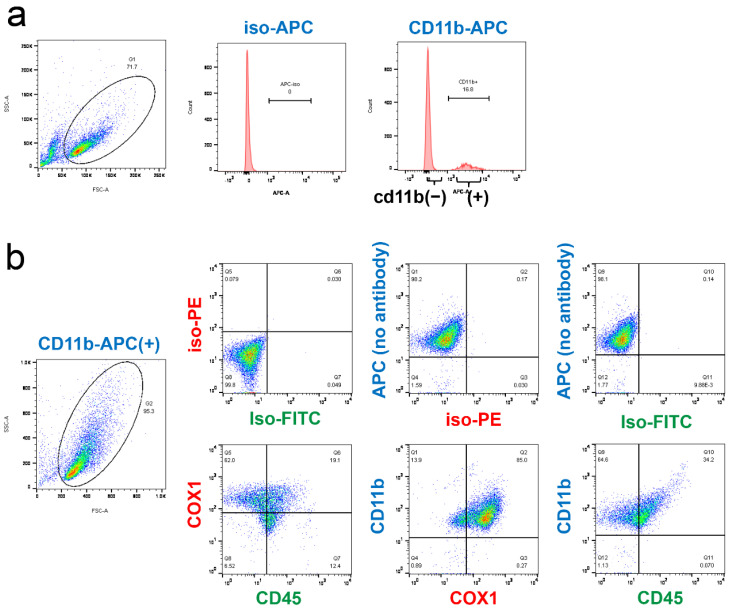
Analysis of CD45 and COX-1 expression in CD11b-positive cells collected from the CNS. (**a**) Cells dissociated from mouse brain were stained with CD11b-APC (right panel) or isotype-control (iso) (middle panel), which was followed by the sorting of APC-positive cells by flow cytometry. (**b**) CD11b-positive cells sorted in (**a**) were stained with CD11b-APC, CD45-FITC, and COX1-PE (lower panels), or with isotype controls for FITC and PE (upper panels). Note that all cells were APC-positive without staining, confirming successful sorting. Expression of COX-1 was shown in the CD45/CD11b-double positive cells of the CNS.

**Figure 5 ijms-22-03396-f005:**
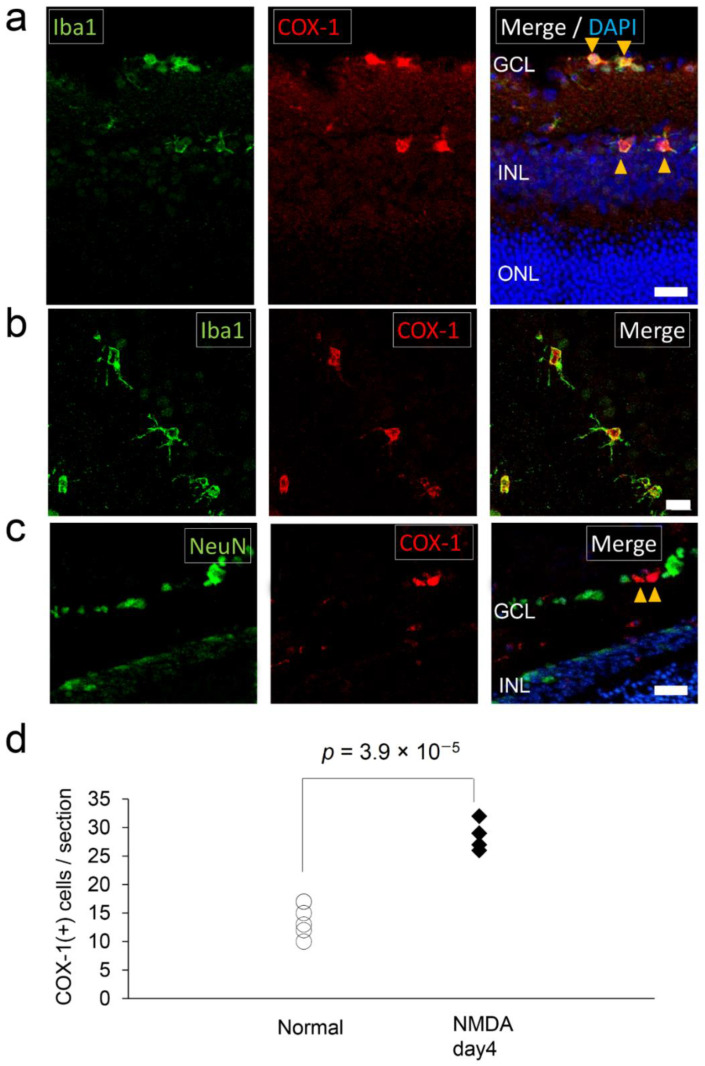
COX-1^+^ microglia/macrophage in the *N*-methyl-D-aspartate (NMDA) retinal neurotoxicity model. NMDA was administered into the vitreous cavity of the mice to induce neurotoxic damage to the inner layer of the retina. (**a**) Photomicrographs of the retina in NMDA-administrated model mice on day four after NMDA-administration labeled with Iba1 (left) and COX-1 (middle). Iba1^+^ microglia/macrophage localized in the GCL and the INL. COX-1 staining coincided with Iba1^+^ microglia/macrophage (yellow arrowheads). Scale bar, 20 μm. (**b**) Photomicrographs showing a magnified image of microglia/macrophage in the retina of NMDA-administrated model mice. The processes of microglia/macrophage were shortened and the cell bodies of microglia/macrophage were larger than their resting state (also see Figure 1b). Scale bar, 10 μm. (**c**) Photomicrographs of the retina in NMDA-administrated model mice labeled with NeuN (left) and COX-1 (middle). COX-1 staining (yellow arrowheads) did not coincide with NeuN^+^ neurons in the GCL and the INL. Scale bar, 20 μm. (**d**) A graph showing the number of COX-1^+^ cells in a retinal section of the NMDA-administrated retinal neurotoxicity model on day four after NMDA administration (NMDA day four) and normal mice (*n* = 5). All COX-1^+^ cells were Iba1 positive. COX-1^+^ cells were significantly increased in the retina of the NMDA-administrated retinal neurotoxicity model.

**Figure 6 ijms-22-03396-f006:**
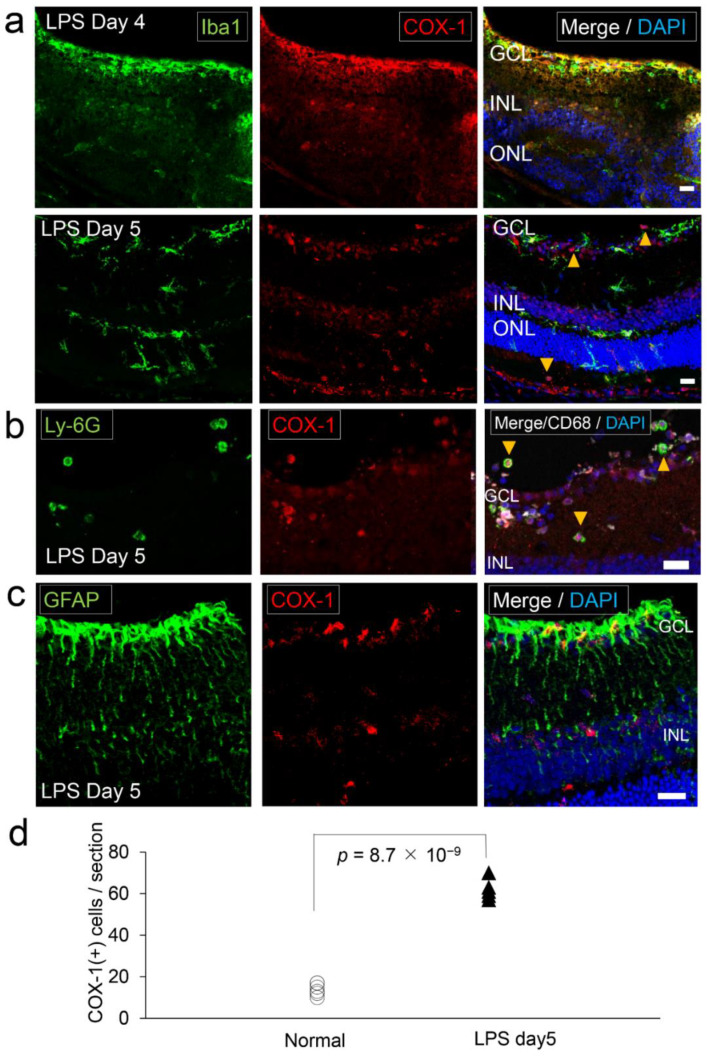
COX-1^+^ microglia/macrophage and neutrophils in the lipopolysaccharide (LPS)-treated ocular inflammation model. LPS was administered into the vitreous cavity of the mice to induce ocular inflammation. Photomicrographs show the retina in LPS-administrated model mice. (**a**) Retinal section labeled with Iba1 (left) and COX-1 (middle). On day four after LPS-administration (LPS day four), COX-1^+^ cells accumulated in the inner layer of the retina near the GCL, most of which coincided with Iba1^+^ staining. On day five, COX-1^+^ cells were present throughout the whole retina, and some COX-1^+^ cells did not coincide with Iba1^+^ staining (yellow arrowheads). Iba1^+^ cells had amoeboid morphology. Scale bar, 20 μm. (**b**) Retinal section (LPS day five) labeled with Ly-6G (neutrophil marker; left) and COX-1 (middle). There were infiltrating Ly-6G^+^ cells. Ly-6G^+^ cells coincided with COX-1^+^ staining (yellow arrowheads). Scale bar, 20 μm. (**c**) Retinal section labeled with GFAP (left) and COX-1 (middle). GFAP immunopositivity was stronger than normal retina (also see Figure 1d), but did not colocalize with COX-1. Scale bar, 20 μm. (**d**) A graph showing the number of COX-1^+^ cells in a retinal section of LPS-treated ocular inflammation model and normal mice (*n* = 5). COX-1^+^ cells include both Iba1^+^ and Iba1^–^ cells. COX-1^+^ cells were significantly increased in the retina of the LPS-treated ocular inflammation model.

**Figure 7 ijms-22-03396-f007:**
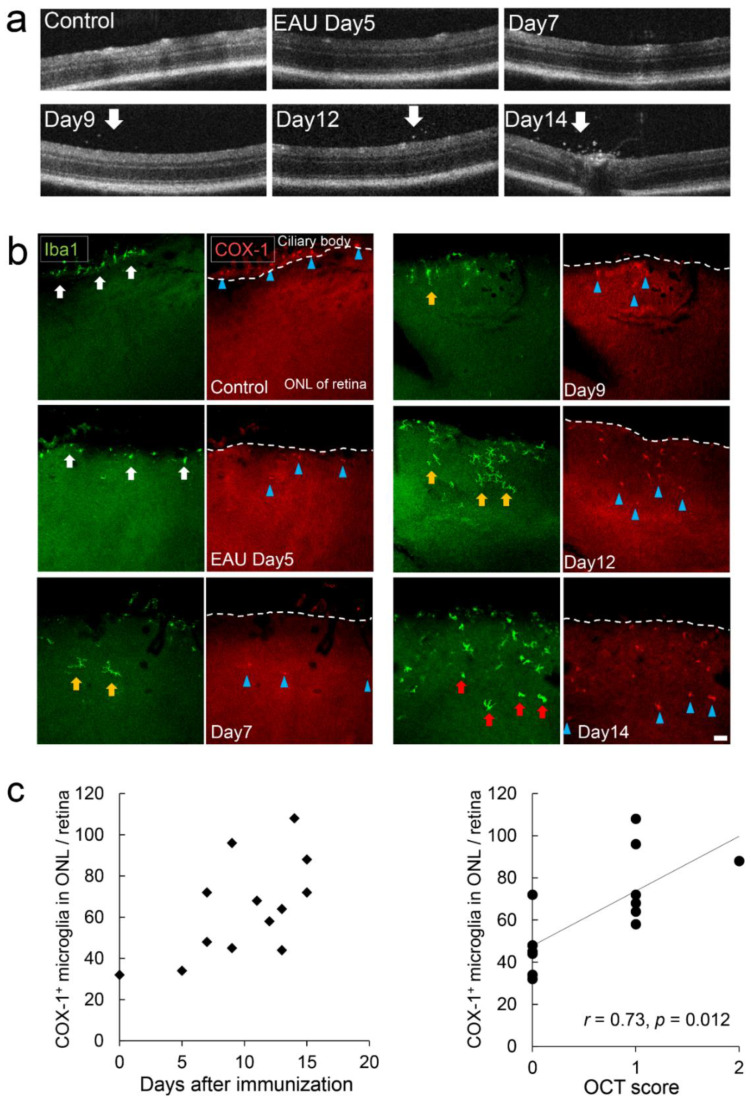
COX-1^+^ microglia/macrophage dynamics in early phase of the experimental autoimmune uveoretinitis (EAU) model. Mice were immunized with IRBP peptide to develop autoimmune uveoretinitis. (**a**) OCT imaging showed cell infiltration into the vitreous (white arrows). Day, the day after immunization. (**b**) Photomicrographs of the flat-mounted retina of EAU mice labeled with Iba1 (left) and COX-1 (right). All COX-1^+^ cells found in the ONL were Iba1^+^ cells. They first had a small cell body and thin processes (white arrows) which became thicker (yellow arrows) and, finally, changed into amoeboid forms (red arrows). Iba1^+^COX-1^+^ cells gradually increased in the ONL of the retina, from the ciliary body side to the center of the retina. Dashed lines show the boundary between the ciliary body and the retina. On days 5, 7, 9, 12, and 14, Iba1^+^COX-1^+^ cells were found as far as 114, 207, 107, 236, and 336 μm from the border to the choroid, respectively (blue arrowheads). Scale bar, 50 μm. (**c**) COX-1^+^ cells infiltrating into the ONL increased with time after immunization. The number of COX-1^+^ cells infiltrating into the ONL correlated with the OCT score. Spearman (*n* = 13, *r* = 0.73, *p* = 0.012).

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
