# Peer review of "Dynamics of Cyclooxygenase-1 Positive Microglia/Macrophage in the Retina of Pathological Model Mice as a Biomarker of the Retinal Inflammatory Diseases"

_ijms, 2021, doi:10.3390/ijms22073396_

Round 1
Reviewer 1 Report
The authors sufficiently addressed the questions/concerns.
Reviewer 2 Report
The authors have responded to my previous critiques, I am pleased to recommend publication of this manuscript.
This manuscript is a resubmission of an earlier submission. The following is a list of the peer review reports and author responses from that submission.
Round 1
Reviewer 1 Report
Makebe et al. presents a manuscript about COX-1 positive retinal microglia as a biomarker of retinal inflammatory diseases. However, the data weakly supported the conclusion.
Major comments:
Fig3: cd11b marker is insufficient to identify microglia or macropahge. Authors could use cd11b(high)/cd45(mid) for microglia and cd11b(high)cd45(high) for macrophage.
Ln181: ....cox-1 is expressed in microglia of the retina and ... It is over-stated. The current data do not directly shown only microglia express cox-1!
Ln194: ....cox-1 can be a biomarker of retinal microglia. It is debatable. This statement may sound in normal/uninjured retina. It is possible that infiltration macrophage also express cox1 in injured retina.
Minor comments:
Results 2.1 and 2.2: It claimed both cox-1 and 2 do not express on other retinal cell types except those shown in the figures. Authors need to show which other retinal cells labeling were used in this study and include the data in the manuscript.
Fig.2a: Need a higher magnification image/insert to show the colocalization of cox-2 and astrocyte. Muller cells body seem weakly expressed cox2. Please clarify.
The purity of retinal microglia and splenic macrophage culture needs to be shown or cite a reference.
All plots showing cox-1 positive cells per section. Does it mean all cox-1+ cells are Iba-1+ in different injured models? In the culture data (Fig3), only 50% Iba1+ cells express cox-1. Please discuss why cox-1 cell number increased after injury and would it be infiltrating macrophage?
Fig3: Please add the plots of isotype control. Show the number of culture has been used in this study and add error bar to the charts.
Fig5a: The image of Iba1+ cells is unclear. Please mark those cox-1+/Iba1- cells and discuss the possible identify of these cells. Note in Result 2.1, it said other retinal cell types (except Iba1+ cells) do not express cox1.
Ln313: The authors sacrificed mice at 5 different time points in EAU model. Detailed description of each time point is missing in Results.
Reviewer 2 Report
This very interesting manuscript from Makabe and co-workers demonstrates that COX-2 expressed in microglia and/or macrophage in the retina could be a potential biomarker of inflammatory conditions of retinal diseases. The experiments in this study seem carefully conducted and the manuscript is well prepared. I have a few comments.
- The authors use the term microglia and macrophage. However, a certain macrophages and microglia express iba-1(microglia/macrophage-specific calcium-binding protein). Therefore, the title should be changed as follow: Dynamics of cyclooxygenase-1 positive microglia/macrophage in the retina of pathological model mice as a biomarker of the retinal inflammatory diseases. Also, microglia should be changed to microglia/macrophage in the text.
- The authors should comment on the morphology of the microglia in Fig 1, 2, 4, 5 and in 3 different animal models.
Are they activated or resting in Fig 4-6 ?
- Do the authors have any thoughts about COX-1 concentration in eyes with 3 different animal models? Maybe it would be useful for biomarker.
- The authors should provide more explanation regarding COX-2 positivity in inner nuclear layer, probably in amacrine, horizontal, and Muller cells in figure legends, results, and discussion.